# LOGARITHMIC LINEAR UNITS (LOGLUS): A NOVEL ACTIVATION FUNCTION FOR IMPROVED CONVERGENCE IN DEEP NEURAL NETWORKS

## ABSTRACT

The Logarithmic Linear Unit (LogLU) presents a novel activation function for deep neural networks by incorporating logarithmic elements into its design, introducing non-linearity that significantly enhances both training efficiency and accuracy. LogLU effectively addresses common limitations associated with widely used activation functions include ReLU, Leaky ReLU, and ELU, which suffer from issues like the dead neuron problem and vanishing gradients. By enabling neurons to remain active with negative inputs and ensuring effective gradient flow during backpropagation, LogLU promotes more efficient convergence in gradient descent. Its capability to solve fundamental yet complex non-linear tasks, such as the XOR problem, with fewer neurons demonstrates its efficiency in capturing non-linear patterns. Extensive evaluations on benchmark datasets like Caltech 101 and Imagenette, using the InceptionV3 architecture, reveal that LogLU not only accelerates convergence but also enhances model performance compared to existing activation functions. These findings underscore LogLU's potential as an effective activation function that improves both model performance and faster convergence.

## 1 INTRODUCTION

Deep learning has become highly popular in recent years for its ability to recognize complex patterns within data. LeCun et al. (2015). At the core of deep learning models are layers of neurons, A neural network processes input data by passing it through layers of weighted connections, where activation functions are applied to determine the output at each node. The choice of activation function is critical, as it influences how well a neural network learns, handles non-linearities, and performs in various tasks Goodfellow et al. (2016). An effective activation function enables the model to approximate complex relationships, This ability is a key reason for deep learning's better performance.

While popular activation function like Rectified Linear Unit (ReLU) Nair & Hinton (2010), Leaky ReLU Xu et al. (2015), and Exponential Linear Unit (ELU) Clevert et al. (2015) have been widely used, they each present limitations. For instance, ReLU faces the dead neuron problem, where neurons stop learning if they constantly receive negative inputs. Although Leaky ReLU addresses this problem by permitting small negative values, it introduces the vanishing gradient problem, limiting its effectiveness in deep networks Maas (2013). ELU, on the other hand, provides a smooth output for negative inputs but increases computational complexity due to its exponential calculation.

In this paper, we introduce a new activation function i.e., Logarithmic Linear Unit (LogLU), which addresses the limitations of existing activation functions. LogLU is designed to solve both the dead neuron and vanishing gradient problems while maintaining computational simplicity. It enables neurons to remain active even with negative inputs, preventing dead neurons and ensuring that gradients stay sufficiently large during backpropagation. This enhances the training of deep neural networks, resulting in quicker convergence and improved overall performance. One of the most notable features of LogLU is its ability to solve classic XOR function approximation problems using a single neuron McCulloch & Pitts (1943). This highlights its efficiency in capturing non-linear relationships with fewer resources compared to other activation functions. LogLU's unique properties allow deep

learning models to perform complex tasks with fewer neurons, making it an optimal choice for both small-scale and large-scale applications. In addition to addressing the dead neuron and vanishing gradient problems, LogLU demonstrates superior computational efficiency. In our experiments, we compare the time complexity of LogLU against popular activation functions, including Swish and Mish Ramachandran et al. (2018), across a variety of benchmark datasets. LogLU consistently outperforms in terms of both training speed and validation accuracy.

The overall analysis of this paper is as follows:

- Introduction of a new activation function, LogLU, that effectively addresses the dead neuron and vanishing gradient and Exploding Gradient problems.
- LogLU has successfully solved the classic XOR problem. This showcases LogLU's efficiency in handling basic logic operations with simplicity.
- LogLU activation was compared with popular activation functions across a range of benchmark datasets, highlighting its superior performance.

## 2 DIFFERENTIABILITY OF THE LOGLU ACTIVATION FUNCTION

### 2.1 DEFINITION OF THE LOGLU FUNCTION

The LogLU activation function is defined as:

$$f(x) = \begin{cases} x, & \text{if } x > 0 \\ -\log(-x+1), & \text{if } x \leq 0 \end{cases} \tag{1}$$

**Case 1: Differentiability for** $x > 0$ For $x > 0$, the function simplifies to:

$$f(x) = x$$

The derivative is:

$$f'(x) = \frac{d}{dx}(x) = 1$$

**Case 2: Differentiability for** $x \leq 0$**:** For $x \leq 0$, the function is:

$$f(x) = -\log(-x+1)$$

To find its derivative, use the chain rule. Let $g(x) = -x + 1$, then:

$$f(x) = -\log(g(x))$$

The derivative of $-\log(g(x))$ with respect to $x$ is:

$$\frac{d}{dx}[-\log(g(x))] = -\frac{1}{g(x)} \cdot \frac{d}{dx}[g(x)] = \frac{1}{1-x}$$

Thus, the function is differentiable for $x \leq 0$ with:

$$f'(x) = \frac{1}{1-x}$$

**Continuity and Differentiability at** $x = 0$ Evaluating the function at $x = 0$:

$$f(0) = -\log(0+1) = -\log(1) = 0$$

The derivative from the right of $x = 0$ (as $x \to 0^+$) is:

$$f'(0^+) = 1$$

The derivative from the left of $x = 0$ (as $x \to 0^-$) is:

$$f'(0^-) = \frac{1}{1 - 0} = 1$$

Since $f'(0^+) = f'(0^-) = 1$, the function is both continuous and differentiable at $x = 0$.

The LogLU activation function is differentiable for all $x$, including at $x = 0$.

## 2.2 NON-LINEARITY OF THE LOGLU ACTIVATION FUNCTION

For $x > 0$: In this domain, the function is $f(x) = x$, which is linear and does not exhibit non-linearity.

For $x \le 0$: In this domain, the function is:

$$f(x) = -\log(-x + 1)$$

To verify non-linearity, compute the second derivative.

**First Derivative:**

$$f'(x) = \frac{1}{1 - x}$$

**Second Derivative:**

Applying the quotient rule:

$$f''(x) = \frac{d}{dx}\left(\frac{1}{1 - x}\right) = \frac{1}{(1 - x)^2}$$

Since the second derivative is non-zero, $f(x) = -\log(-x + 1)$ is non-linear for $x \le 0$. The LogLU activation function introduces non-linearity for negative inputs, which is essential for modeling complex functions in neural networks.

## 2.3 MITIGATION OF THE VANISHING AND EXPLODING GRADIENT PROBLEMS

For $x > 0$: The gradient is:
$$f'(x) = 1$$

This constant and bounded gradient in the positive domain prevents both the vanishing and exploding gradient problems.

For $x \le 0$: The gradient is:
$$f'(x) = \frac{1}{1 - x}$$

As $x \to 0$, $f'(x) \to 1$, and as $x \to -\infty$, $f'(x) \to 0$. Although the gradient decreases for large negative values, it remains non-zero, mitigating the vanishing gradient problem. Furthermore, since the gradient is bounded and decreases for negative values, it avoids the exploding gradient problem. The LogLU activation function effectively mitigates the vanishing gradient problem by maintaining a non-zero gradient for negative inputs and a constant gradient for positive inputs, while also avoiding the exploding gradient problem due to its bounded gradient across all input values.

## 3 LEARNING XOR FUNCTION WITH LOGLU

The XOR function complex operation commonly used to analyze the performance of activation functions in neural networks. The architecture of the neural network designed to model the XOR function, as shown in Figure 1, The network consists of three neurons in the hidden layer and one output neuron. The hidden layer uses the LogLU activation function, while the output layer employs the sigmoid activation function. This combination allows the network to effectively capture the non-linearity of the XOR function while ensuring stable output scaling.

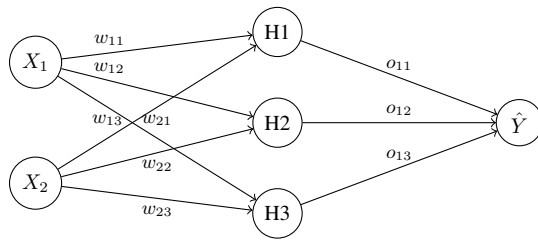

Figure 1: Neural Network Architecture for XOR Function for LogLU as Activation Function.

The network undergoes both feedforward Hornik et al. (1989) and backpropagation processes Rumelhart et al. (1986) during training. In the feedforward pass, the inputs are propagated through the network to generate a prediction. The weights and biases in the network are adjusted using the backpropagation algorithm to minimize the error between the actual and predicted outputs. By iteratively updating the weights through this process, the network learns to approximate the XOR function accurately.

### 3.1 FORWARD PROPAGATION FOR XOR FUNCTION

The forward Propagation through the neural network involves computing the activations for the hidden layer neurons using the LogLU activation function is applied to the summation of the result obtained by multiplying the inputs and weights using the dot product and their respective biases Hornik et al. (1989) as shown in Table 1. Specifically, for each hidden neuron $H_i$, the activation is given by:

$$H_i = \text{LogLU}(w_{i1}X_1 + w_{i2}X_2 + ..... + b_i)$$

where $w_{ij}$ are the weights and $b_i$ is the bias for neuron $H_i$ Cybenko (1989).

Following the computation of the hidden layer activations, the output neuron activation is calculated using the sigmoid activation function. The activation for the output neuron $\hat{Y}$ is given by:

$$\hat{Y} = \sigma\left(o_{11}H_1 + o_{12}H_2 + o_{13}H_3 + ..... + b_{\text{out}}\right)$$

where $o_{ij}$ are the weights from the hidden layer to the output neuron, $H_i$ are the activations from the hidden layer, and $b_{\text{out}}$ is the bias for the output neuron Hornik et al. (1989).

### 3.2 BACKPROPAGATION FOR XOR FUNCTION

During the backpropagation process, The objective is to minimize the error between the predicted outputs and the actual values by adjusting the weights and biases. By applying the chain rule Rumelhart et al. (1986) to compute the gradients of the loss function with respect to each weight and bias in the neural network.

**Output Layer**: Specifically, for each output weight $o_{ij}$, the gradient is given by:

$$\frac{\partial L}{\partial o_{ij}} = \frac{\partial L}{\partial \hat{Y}} \cdot \frac{\partial \hat{Y}}{\partial o_{ij}} = \delta_{\hat{Y}} \cdot H_i$$

where $L$ is the loss function, $\delta_{\hat{Y}} = \hat{Y} - Y$ Bishop (2006) is the error at the output layer (for sigmoid) Han & Moraga (1995), and $H_i$ is the activation from the hidden layer. The weights are updated using:

$$o_{ij}^{(t+1)} = o_{ij}^{(t)} - \eta \cdot \frac{\partial L}{\partial o_{ij}}$$

where $\eta$ is the learning rate.

**Hidden Layer**: Next, we compute the gradients for the hidden layer neurons Bengio (2009). The gradient of the loss with respect to the activation $H_i$ is given by:

$$\frac{\partial L}{\partial H_i} = \sum_j \frac{\partial L}{\partial \hat{Y}} \cdot \sigma'(\hat{Y}) \cdot o_{ij}$$

The gradient with respect to the weights $w_{ij}$ is computed as:

$$\frac{\partial L}{\partial w_{ij}} = \frac{\partial L}{\partial H_i} \cdot \text{LogLU}'(H_i) \cdot X_j$$

where $\text{LogLU}'(H_i)$ is the derivative of the LogLU activation function as shown in Table 1.

Table 1: Weights, biases, and predictions for the XOR logic gate using LogLU.

| Input $(X_1, X_2)$ | Actual Output / Thresholded Prediction | Hidden Layer Weights | Output Layer Weights |
|---|---|---|---|
| $(0,0)$ | 0 / 0 | $w_{11} = 0.7,\ w_{21} = -0.6$ $w_{12} = 0.5,\ w_{22} = 0.4$ $w_{13} = -0.1,\ w_{23} = 0.7$ | |
| | | | $o_{11} = 1.2,\ o_{12} = -0.6,\ o_{13} = 0.1$ |
| $(0,1)$ | 1 / 1 | $w_{11} = 0.7,\ w_{21} = -0.6$ $w_{12} = 0.5,\ w_{22} = 0.4$ $w_{13} = -0.1,\ w_{23} = 0.7$ | |
| | | | $o_{11} = 1.2,\ o_{12} = -0.6,\ o_{13} = 0.1$ |
| $(1,0)$ | 1 / 1 | $w_{11} = 0.7,\ w_{21} = -0.6$ $w_{12} = 0.5,\ w_{22} = 0.4$ $w_{13} = -0.1,\ w_{23} = 0.7$ | |
| | | | $o_{11} = 1.2,\ o_{12} = -0.6,\ o_{13} = 0.1$ |
| $(1,1)$ | 0 / 0 | $w_{11} = 0.7,\ w_{21} = -0.6$ $w_{12} = 0.5,\ w_{22} = 0.4$ $w_{13} = -0.1,\ w_{23} = 0.7$ | |
| | | | $o_{11} = 1.2,\ o_{12} = -0.6,\ o_{13} = 0.1$ |

## 4 COMPUTATIONAL TIME COMPLEXITY FOR ACTIVATION FUNCTIONS

The computational time complexity of activation functions plays a crucial role in determining their efficiency within neural networks. In this study, we assess the execution times of various activation functions by averaging their performance over 10,000 independent runs. Each run involves applying the activation function to a vector of length $10^6$, with elements uniformly distributed in the range $[-10, 10]$, as shown in Figure 2. The results demonstrate that LogLU offers superior computational efficiency compared to other activation functions. Furthermore, the graphical representations of the activation functions, along with their respective first derivatives, are illustrated in Figure 3. All corresponding activation function formulas are provided in detail in Table 2.

LogLU operates linearly for positive values, passing through unchanged, and smoothly transitions near zero as $-\log(-x + 1)$ approaches zero for slightly negative values Figure 3. As $x$ becomes more negative, LogLU grows logarithmically, resulting in a slower increase in magnitude compared to Leaky ReLU Xu et al. (2015) or ELU Clevert et al. (2015), which scale negative inputs linearly or exponentially. This smooth transition and bounded behavior for negative values give LogLU a unique advantage over ReLU Nair & Hinton (2010), as it retains sensitivity to negative inputs and

Table 2: Activation Functions and Their Equations

| Activation Function | Equation |
|---|---|
| ReLU | $f(x) = \max(0, x)$ |
| Leaky ReLU | $f(x) = \begin{cases} x, & \text{if } x > 0 \\ \alpha x, & \text{if } x \leq 0 \end{cases}$ |
| ELU | $f(x) = \begin{cases} x, & \text{if } x > 0 \\ \alpha(\exp(x) - 1), & \text{if } x \leq 0 \end{cases}$ |
| Sigmoid | $f(x) = \frac{1}{1+\exp(-x)}$ |
| Tanh | $f(x) = \frac{\exp(x)-\exp(-x)}{\exp(x)+\exp(-x)}$ |
| Swish | $f(x) = x \cdot \sigma(x)$ where $\sigma(x) = \frac{1}{1+\exp(-x)}$ |
| Mish | $f(x) = x \cdot \tanh(\text{softplus}(x))$ where $\text{softplus}(x) = \log(1 + \exp(x))$ |
| **LogLU** | $f(x) = \begin{cases} x, & \text{if } x > 0 \\ -\log(-x + 1), & \text{if } x \leq 0 \end{cases}$ |

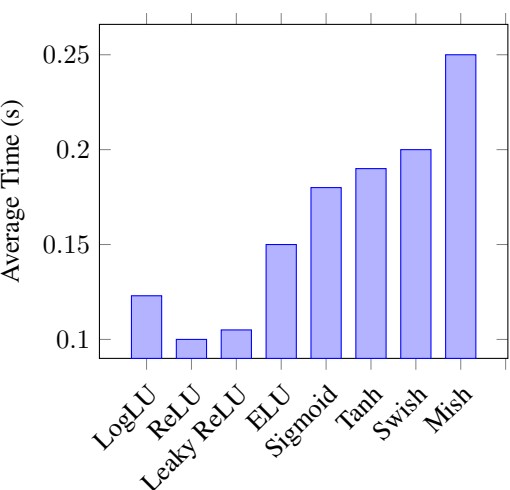

Figure 2: Bar Graph of Average Computation Times (s) for Various Activation Functions

captures more features. Compared to Leaky ReLU, LogLU's slower, logarithmic growth results in more controlled activations, potentially enhancing gradient flow and stability. Additionally, LogLU decays faster than Mish Ramachandran et al. (2018) for negative values, offering a more conservative and efficient approach to managing negative activations, reducing the risk of exploding gradients and aiding in model convergence Maas (2013).

In the derivative graph, the derivative of LogLU for negative values decreases smoothly, facilitating some gradient flow unlike ReLU Nair & Hinton (2010), while not maintaining the constancy observed in Leaky ReLU Xu et al. (2015). This characteristic can help mitigate issues such as dead neurons in ReLU while providing a controlled response to negative inputs. In contrast, Mish allows even more negative information to pass through than both Swish Ramachandran et al. (2018) and LogLU. Its gradient decays more slowly than that of Swish and significantly slower than LogLU. While Swish exhibits a smooth decrease in gradient, it does so much more gradually compared to the sharper decay seen in LogLU Figure 3.

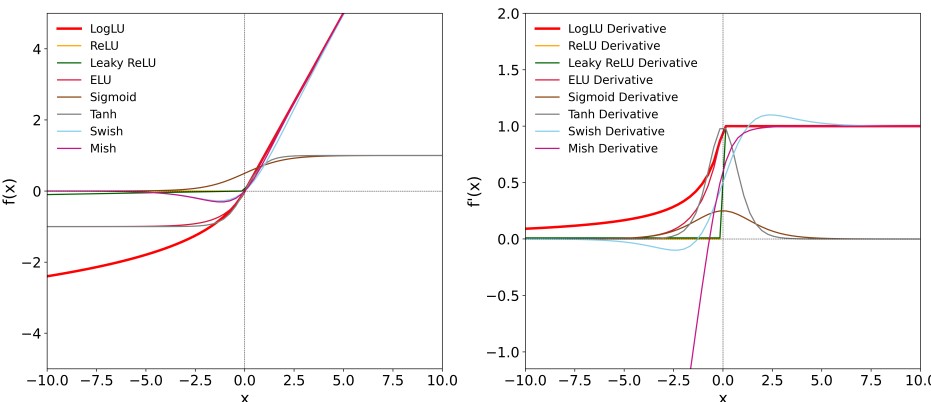

Figure 3: Graphical Comparison of equations Activation Function Curves (Left) vs. First Derivative Curves (Right)

## 5 PERFORMANCE EVALUATION ON BENCHMARK DATASETS

We evaluated benchmark datasets to compare activation functions using the Caltech 101 and Imagenette datasets. The Caltech 101 Fei-Fei et al. (2004) dataset contains 9,144 images, split into 7,280 for training and 1,864 for validation across 101 classes. The Imagenette dataset Howard (2019), a subset of ImageNet with 10 classes, includes 13,394 images, with 9,469 for training and 3,925 for validation. We used the Adadelta optimizer Zeiler (2012) with learning rate of 0.01 and categorical crossentropy Goodfellow et al. (2016) loss function for both datasets, with a softmax activation function in the output layer. The InceptionV3 model Szegedy et al. (2016) was trained on both datasets, utilizing pretrained ImageNet weights PyTorch (2024). The model has over 73M parameters for Caltech 101 and 37M parameters for Imagenette. Training was conducted for 30 epochs on Caltech 101 and 20 epochs on Imagenette, allowing for a comprehensive comparison of activation functions.

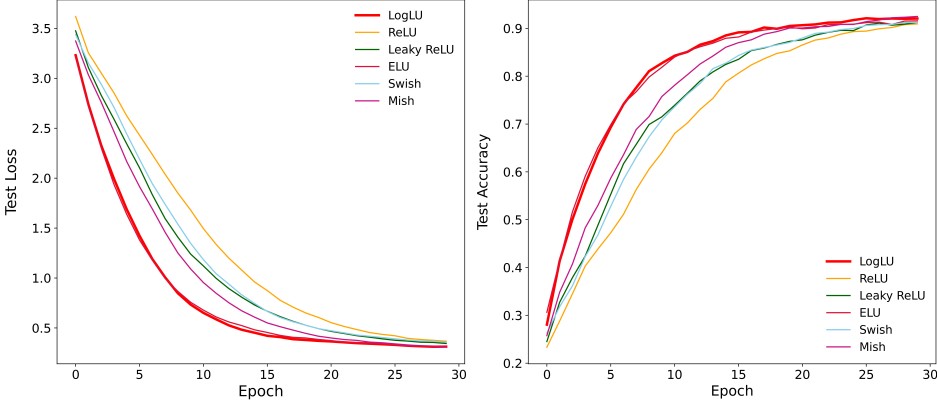

Figure 4: Test Dataset Loss (Left) and Accuracy (Right) on the Caltech 101 Dataset

The results from both the Caltech 101 dataset, illustrated in Figure 4, and the Imagenette dataset, presented in Figure 5, show consistent trends in model performance across various activation functions. The LogLU activation function significantly improved gradient convergence during training, leading to enhanced overall performance. Compared to traditional activation functions like ReLU Nair & Hinton (2010) and Leaky ReLU Xu et al. (2015), LogLU exhibited faster convergence and greater stability, resulting in improved accuracy and reduced loss values. Specifically, as shown in Table 3, LogLU enhances the model's performance to generalize and accurately predict outcomes in the Caltech 101 dataset. Similarly, Table 4 and Figure 5 indicate that LogLU demonstrates improved performance on the Imagenette dataset, achieving higher accuracy and lower loss. These findings suggest that LogLU enhances generalization and learning efficiency across diverse datasets, making

it a valuable tool for optimizing neural network performance in image classification by accelerating convergence and improving model accuracy.

Table 3: Performance Evaluation of Activation Functions on the Caltech 101 Dataset

| Dense Layers | Accuracy % | Val Accuracy % | Loss | Val Loss |
|---|---|---|---|---|
| ReLU | 84.50 | 90.93 | 0.6236 | 0.3674 |
| Leaky ReLU | 86.42 | 91.26 | 0.5183 | 0.3441 |
| ELU | 89.82 | 91.58 | 0.3832 | 0.3070 |
| Swish | 87.55 | 91.26 | 0.4777 | 0.3570 |
| Mish | 87.72 | 92.49 | 0.4679 | 0.3100 |
| **LogLU** | 90.12 | 92.06 | 0.3839 | 0.3126 |

Table 4: Performance Evaluation of Activation Functions on the Imagenette Dataset

| Dense Layers | Accuracy % | Val Accuracy % | Loss | Val Loss |
|---|---|---|---|---|
| ReLU | 91.41 | 94.19 | 0.2774 | 0.1770 |
| Leaky ReLU | 91.61 | 94.11 | 0.2719 | 0.1772 |
| ELU | 91.47 | 94.04 | 0.2640 | 0.1779 |
| Swish | 91.86 | 94.37 | 0.2640 | 0.1742 |
| Mish | 90.86 | 94.39 | 0.2707 | 0.1780 |
| **LogLU** | 91.71 | 94.47 | 0.2518 | 0.1761 |

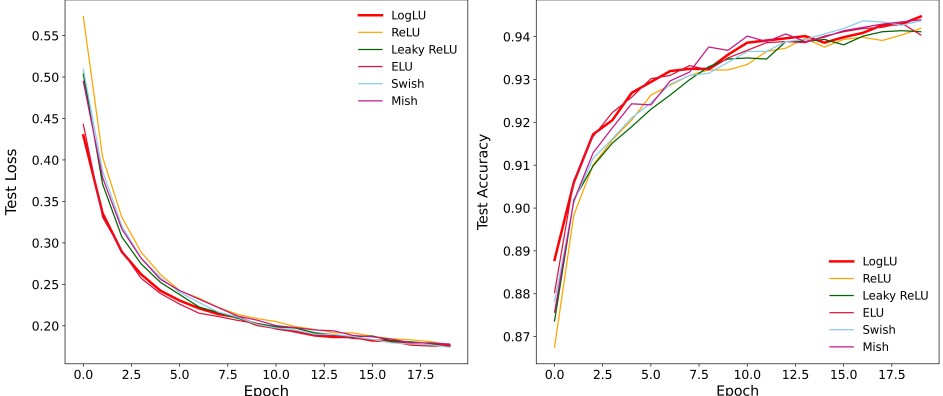

Figure 5: Test Dataset Loss (Left) and Accuracy (Right) on the Imagenette Dataset

## 6 CONCLUSION

This research study focuses on evaluating the impact of various nonlinear activation functions on the performance of output neurons in deep learning models. We specifically examine the performance of well-established activation functions, including ReLU, Leaky ReLU, and ELU, each of which presents certain limitations. To overcome these challenges, we introduce a novel activation function, the Logarithmic Linear Unit (LogLU), designed to enhance the efficiency of model training. The ability of LogLU to keep neurons active with negative inputs and maintain robust gradient flow during backpropagation enables more efficient convergence in gradient descent, particularly in solving complex non-linear tasks. Through extensive evaluations on benchmark datasets, including Caltech 101 and Imagenette, which are relevant for large-scale applications, we demonstrate that LogLU accelerates convergence and improves model performance when integrated into the InceptionV3 architecture. The empirical results show that LogLU consistently outperforms traditional activation functions in terms of convergence speed, stability, accuracy, and loss reduction.

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
