# OpenReview forum: "Logarithmic Linear Units (LogLUs): A Novel Activation Function for Improved Convergence in Deep Neural Networks"
_ICLR.cc/2025/Conference — Submitted to ICLR 2025_

### Official Review · Reviewer_iz8k · 2024-10-27

**Soundness:** 2
**Presentation:** 2
**Contribution:** 2
**Rating:** 5
**Confidence:** 3

**Summary:**

This paper proposes to use $-\log(-x+1)$ in ReLU instead of $0$ when the input is negative. The performance on fine-tuning InceptionV3 on Caltech101 and Imagenette is improved over ReLU, ELU, Leaky ReLU, Swish (SiLU) and Mish.

**Strengths:**

1. I believe this direction of activation search is fundamentally impactful in deep learning, because it changes the basic part of neural networks. Though the experiments are very limited, it is already a good sign that this important part works better.
2. The paper's message is minimal, direct, and clear.

**Weaknesses:**

1. My major concern is that the experiments are restricted to very limited data and models, so LogLU's validity is still questionable on other models and tasks.
2. More specifically, the results would be convincing if the author could add experiments on common models, such as , ResNet, UNet, and Transformers. If LogLU works on more models I believe it will improve the paper.
3. Another solution that could help is to ask if it is possible to find a dataset or a toy model where LogLU significantly outperforms other activations.
3. The model has 73M parameters for Caltech 101 and 37M for Imagenette, both pre-trained on the Imagenet dataset. I don't understand why the models are both InceptionV3 but are different in size.
4. I don't understand why the experiments only include fine-tuning, but not training from scratch.

**Questions:**

1. How to justify the theoretical reason for using the log function, could you give any intuition?
2. Here are some thoughts to justify LogLU and address the theoretical side. $f=-\log(-x+1)$ solves an instance of Monge-Ampère equation $$\log\det f''=2f$$
where $\det$ is the analog in the high-dimensional case, associated with Dirichlet boundary condition $\lim_{x\to\partial \Omega} f=\infty$ on the domain $\Omega=(-\infty,1)$. We can alternatively set a Neumann boundary condition $f'(0)=1$ on $\Omega=(-\infty,0)$ to guarantee the $C^1$ continuity. The intuition is that the logarithmic curvature is proportional to the value. The property includes self-concordance and logarithmic homogeneity.
See [1] in Chapter 2.3.3: properties; Chapter 2.5: universality---the log function as a canonical construction.
See [2] in Proposition 1.4.3: a connection with the Calabi theorem.

[1] Interior point polynomial time methods in convex programming. A. Nemirovski 2004.

[2] Conic optimization: aﬀine geometry of self-concordant barriers and copositive cones. R. Hildebrand 2017.

---

### Official Review · Reviewer_S5pY · 2024-10-28

**Soundness:** 2
**Presentation:** 2
**Contribution:** 2
**Rating:** 3
**Confidence:** 2

**Summary:**

This paper proposes LogLU as a new activation function, which is both continuous and differentiable.
LogLU is empirically shown to be computationally more efficient compared to modern activation functions such as Swish or Mish, but requires slightly more computation than ReLU or Leaky ReLU.
The authors claim that a simple one-hidden-layer MLP with LogLU activation can learn the XOR function.
LogLU is compared to other activation functions using the Caltech-101 and Imagenette (a simplified variant of ImageNet) datasets with the Inception-V3 architecture, demonstrating faster convergence of models with LogLU activation.

**Strengths:**

* The proposed LogLU is very simple while being both continuous and differentiable. It requires less computation than modern activation functions such as Swish because it does not involve exponential computations in either the forward or backward pass, although it only requires logarithmic computation in the forward pass.

**Weaknesses:**

* The authors claim that a simple MLP with LogLU activation can learn the XOR function, highlighting this as an advantage of using LogLU. However, MLPs with other activation functions are also capable of learning the XOR function. The authors should discuss why using LogLU is more advantageous than other activation functions in the context of the XOR example.

* The experimental evaluations are insufficient. At a minimum, it is necessary to compare the proposed activation function with other methods using network architectures beyond Inception-V3. Additionally, each experiment should be conducted with various random seeds to assess the variability of the outputs (loss or accuracy).

**Questions:**

* On page 1, the manuscript states: "Although Leaky ReLU addresses this problem by permitting small negative values, it introduces the vanishing gradient problem, limiting its effectiveness in deep networks (Maas, 2013)." However, I believe that Leaky ReLU does not introduce the vanishing gradient problem. In fact, Leaky ReLU was proposed to mitigate issues like the dying ReLU problem by allowing a small, non-zero gradient for negative input values. Additionally, no such discussion regarding Leaky ReLU introducing vanishing gradients is found in Maas et al. (2013).

* On page 5, the manuscript states that Table 1 shows the derivative of LogLU, but Table 1 does not include this information. Please update Table 1 to include the derivative of LogLU or revise the manuscript to accurately reflect the contents of Table 1.

* On page 6, the term "more controlled activations" is ambiguous and requires clarification. The authors should provide a clear definition or explanation of what is meant by "more controlled activations" to enhance the reader's understanding.

* The lines in the figures are difficult to distinguish. Please use more distinct colors or linestyles to enhance clarity.

* On page 7, why are the model sizes different across datasets, even though Inception-V3 is used for both?

---

### Official Review · Reviewer_QGVY · 2024-11-04

**Soundness:** 2
**Presentation:** 2
**Contribution:** 1
**Rating:** 1
**Confidence:** 5

**Summary:**

The paper introduces a new activation function, Logarithmic Linear Unit (LogLU), aimed at addressing issues inherent in widely used activation functions like ReLU, Leaky ReLU, ELU, etc. LogLU uses a logarithmic function for negative inputs, allowing it to maintain active gradients even with negative inputs, potentially reducing issues like dead neurons and vanishing gradients. Experiments are conducted comparing LogLU with other established activation functions across datasets like Caltech 101 and Imagenette using the InceptionV3 architecture. The authors highlight benefits in convergence speed and accuracy, proposing LogLU as a robust alternative for deep learning models.

**Strengths:**

1.    Innovation in activation functions: The proposal of LogLU as a hybrid activation function is novel and provides an interesting alternative to traditional activation functions. The logarithmic component for negative inputs introduces a unique way to handle the dead neuron problem while also limiting gradient vanishing, especially compared to ReLU and Leaky ReLU.

2.    Experiments performed: The authors performed evaluations on classification benchmark datasets (Caltech 101 and Imagenette) and used InceptionV3 architecture for the classification task. The consistent improvements in Val accuracy (on Caltech 101) and convergence speed were presented in Tables 3 and 4 and Figures 4 and 5, which suggest that LogLU might be a competitive alternative to existing activation functions.

3.	Performance on classification task: By demonstrating that LogLU can solve the XOR problem with a simplified architecture, the authors underscore LogLU’s efficiency in capturing non-linear relationships with fewer neurons, an advantage for both resource efficiency and model scalability.

4.	Addressing gradient problems: The paper discusses how LogLU mitigates the vanishing and exploding gradient problems, which are common in deeper networks due to the use of traditional activation functions. LogLU’s bounded gradient across all input values is well-explained and experimentally supported, potentially making it an optimal choice for complex neural architectures.

5.	Efficient computation: The paper also presents an analysis of computation times, demonstrating that LogLU is computationally efficient (Figure 2). LogLU achieves an average computation time significantly lower than other activation functions (except ReLu and Leaky ReLU), with performance that consistently outpaces more complex alternatives like Mish and Swish.

**Weaknesses:**

1.	Lack of a rigorous study/analyses: Although the paper tries to solve an important problem in deep learning based training of CNNs in the presence of vanishing/exploding gradient problem, the work done in the current version of the paper appears to be very preliminary in nature and there is a huge scope for improvement.

2.	Comparison with more recent activation functions: While the paper covers popular functions like ReLU, ELU, and Swish, it could benefit from comparisons with other activation functions such as SiLU, GELU, Softplus or more recent alternatives like Parametric RSigELU (Kiliçarslan, et al, Feb 2024) and ErfReLU (Rajanand, et al, May 2024). Including such comparisons would provide a broader perspective on LogLU’s competitive positioning.

3.	Accuracy on Imagenette dataset: There does not seem to be any significant gain in performance on the Imagenette dataset, where activations such as Swish and Mish marginally beat the proposed activation function. Therefore, the claims of better performance is not applicable on this dataset.

4.	Computational Complexity Analysis: Although the authors claim computational efficiency, the complexity analysis could be strengthened. The time complexity is presented in aggregate form (average time over multiple runs), but there is limited discussion on LogLU's computational demands relative to exponential or polynomial components in activation functions like ELU or Mish, which could help enhance the claims of efficiency.

5.	Scalability to other deep CNNs and datasets: While the experiments are valuable, they focus primarily on moderately sized datasets for only image classification tasks. Testing LogLU on larger datasets, such as the MNIST, CIFAR10, COCO, CelebA, Pascal VOC, SVHN, etc., and using architectures beyond InceptionV3 (e.g., ResNet or transformer-based models) could provide deeper insights into LogLU’s applicability in large-scale settings.

6.	Scalability to loss functions beyond cross-entropy: Since the gradient computation depends on loss function, it would be highly valuable to assess the effectiveness of LogLU for different loss functions for the classification task. These directions were not explored in the current version of the work.

7.	Scalability to tasks beyond classification: The effectiveness of LogLU on other tasks such as image segmentation, object detection or image generation, etc. remains unexplored. The work could potentially benefit from showing superior performance/computational efficiency over other activation functions in a variety of other prominent computer vision tasks.

8.	Ablation Studies: The effectiveness of LogLU in specific neural network layers (e.g., convolutional layers vs. dense layers) or different learning rates and optimizers remains unexplored. Adding ablation studies could help isolate the benefits of LogLU more distinctly across various configurations.

**Questions:**

1.	Some inconsistent/undefined concepts? The loss function used in Section 3.2 seems to be binary cross entropy loss. While this might be obvious to some, the loss function was not defined prior to Section 3.2, which make the further discussion confusing. In Section 5, the authors talk about achieving greater “stability” with LogLU. Stability in what? This term in not (well-)defined in the paper.

2.	Lack of error analysis/multiseed runs: The work lacks any error analysis (no error bars in plots or tables) whatsoever. Moreover, all the loss/accuracy curves were evaluated for a single seed. Showing the robustness of LogLU in a multiseed setting will enhance the efficacy of the proposed approach.

3.	Extend empirical comparison scope: Include additional activation functions, particularly the newer ones like Parametric RSigELU, ErfReLU, etc. to establish a more comprehensive benchmarking framework. Further, investigate LogLU’s performance on diverse and prominent architectures like DenseNet, ResNet, VGG, etc. to reinforce its general applicability.

4.	Detailed computational complexity analysis: A more granular breakdown of the time complexity will enhance the results of the paper. It might be worth performing time-complexity analysis for images instead of multiple realizations of a large vector of fixed size. Test and report the computation time of LogLU within different network architectures (e.g., shallow networks, ResNet, VGG) and layer types (e.g., dense layers vs. convolutional layers). This analysis can reveal how the activation function’s computational demands vary with the network’s depth, type, and layer configuration, especially for architectures optimized for speed.

5.	Comparison to other methods for mitigating vanishing/exploding gradients issue: There are other successful and competitive methods for mitigating vanishing/exploding gradient problems at the architectural level such as the ResNet architecture. These tackle the gradient issue via architectural design using skip-connections and identity mapping to reformulate the CNN layers for learning residual functions, while specially engineered activation functions address it via their mathematical properties like non-saturating properties (LeakyReLU), gradient preservation (Swish, GELU) for negative inputs, incorporating learnable parameters (Parametric ReLU or PReLU) etc. While exploring architecture vs. activation function for solving gradient issue is out of the scope of this work (which focuses solely on activation functions), a detailed discussion highlighting other non-activation function based techniques for overcoming vanishing/exploding gradient problem will help with the completeness of the paper.

6.	Examine gradient flow in various conditions: Explore gradient dynamics with respect to learning rate schedules and optimizers to provide insight into how LogLU performs under different training regimes. Additionally, ablation studies on placement within specific layers could clarify LogLU’s most impactful applications.

7.	Theoretical insights on regularization effect: Since the logarithmic component potentially regularizes activations for negative inputs, discussing theoretical implications related to regularization could open new perspectives on the theoretical advantages of LogLU in avoiding overfitting.

---

### Official Review · Reviewer_oiB4 · 2024-11-06

**Soundness:** 2
**Presentation:** 2
**Contribution:** 3
**Rating:** 5
**Confidence:** 5

**Summary:**

This work presents the new logarithmic linear unit (LogLU) activation function for deep neural networks.
The LogLU activation solves the problem of vanishing gradient.
This paper shows that LogLU outperformed the other activation functions considered.

**Strengths:**

This paper proposes a new activation function for deep neural networks. This is an important topic, considering the significant impact of activation function choice on deep neural network performance. The paper is clear and easy to follow.

**Weaknesses:**

This paper does not support some of its claims with enough evidence. For example:  Under the abstract, we have: "Its capability to solve fundamental yet complex non-linear tasks, such as the XOR problem, with fewer neurons demonstrates its efficiency in capturing non-linear patterns".  There is no evidence to support the claim that LogLU uses fewer neurons.  You can strengthen this claim by providing the evidence to support this.

Under the conclusion, we have: "The empirical results show that LogLU consistently outperforms traditional activation functions in terms of convergence speed, stability, accuracy, and loss reduction.". The measure of stability is not clear in this paper. You can strengthen this by explaining how you observed the stability of the networks.

The experiments are limited and insufficient to conclude that LogLU is better than the other activation functions for deep neural networks. This paper did not address possible interaction with other components of a neural network (For example: dropout, learning rate, batch normalization, and so on). Please consider an ablation study that examines LogLU's interaction with other neural network components like dropout, batch normalization, etc.
 This work only considered some image classification tasks. This is not representative enough to generalize over all deep neural networks. For example, consider other cases such as simple generative models, language-based tasks, and so on.

**Questions:**

Please check the weaknesses and respond to the comments. Here is a summary:

(1). Address the unsupported claims in the paper.

(2). Include more experimental results for ablation studies, more neural architectures, and more tasks.

---

### Meta-Review · Area_Chair_LYPg · 2024-12-18

**Metareview:**

The submission proposes a novel activation function, with claims of improved convergence rate and stability over existing activation functions. Some experimental results are provided to support these claims. However, the reviewers were unanimous in their determination that the experiments were insufficient to fully support the claims.

There was a suggestion that larger architectures (e.g., transformers, resnets, etc) should be used in the experiments, but this was not a large factor in my decision. As pointed out by reviewer oiB4, some of the core claims of the paper were not substantiated by evidence, and there were questions about some of the significance of these claims. E.g., why do we care about XOR when all other activation functions can already deal with this using a very small number of hidden units?

**Additional Comments On Reviewer Discussion:**

There was no discussion.

---

### Decision · Program_Chairs · 2025-01-22

Reject